# Blocking Store-Operated Ca^2+^ Entry to Protect HL-1 Cardiomyocytes from Epirubicin-Induced Cardiotoxicity

**DOI:** 10.3390/cells12050723

**Published:** 2023-02-24

**Authors:** Xian Liu, Yan Chang, Sangyong Choi, Chuanxi Cai, Xiaoli Zhang, Zui Pan

**Affiliations:** 1Department of Kinesiology, College of Nursing and Health Innovation, The University of Texas at Arlington, Arlington, TX 76010, USA; 2Department of Graduate Nursing, College of Nursing and Health Innovation, The University of Texas at Arlington, Arlington, TX 76010, USA; 3Bone and Muscle Research Center, College of Nursing and Health Innovation, The University of Texas at Arlington, Arlington, TX 76010, USA; 4Department of Surgery, Division of Surgical Sciences, School of Medicine, University of Virginia, Charlottesville, VA 22903, USA; 5Department of Biomedical Informatics, The Ohio State University, Columbus, OH 43210, USA

**Keywords:** anthracycline, chemotherapy, store-operated Ca^2+^ entry (SOCE), apoptosis, cardiac hypertrophy, NFAT4, reactive oxygen species (ROS)

## Abstract

Epirubicin (EPI) is one of the most widely used anthracycline chemotherapy drugs, yet its cardiotoxicity severely limits its clinical application. Altered intracellular Ca^2+^ homeostasis has been shown to contribute to EPI-induced cell death and hypertrophy in the heart. While store-operated Ca^2+^ entry (SOCE) has recently been linked with cardiac hypertrophy and heart failure, its role in EPI-induced cardiotoxicity remains unknown. Using a publicly available RNA-seq dataset of human iPSC-derived cardiomyocytes, gene analysis showed that cells treated with 2 µM EPI for 48 h had significantly reduced expression of SOCE machinery genes, e.g., Orai1, Orai3, TRPC3, TRPC4, Stim1, and Stim2. Using HL-1, a cardiomyocyte cell line derived from adult mouse atria, and Fura-2, a ratiometric Ca^2+^ fluorescent dye, this study confirmed that SOCE was indeed significantly reduced in HL-1 cells treated with EPI for 6 h or longer. However, HL-1 cells presented increased SOCE as well as increased reactive oxygen species (ROS) production at 30 min after EPI treatment. EPI-induced apoptosis was evidenced by disruption of F-actin and increased cleavage of caspase-3 protein. The HL-1 cells that survived to 24 h after EPI treatment demonstrated enlarged cell sizes, up-regulated expression of brain natriuretic peptide (a hypertrophy marker), and increased NFAT4 nuclear translocation. Treatment by BTP2, a known SOCE blocker, decreased the initial EPI-enhanced SOCE, rescued HL-1 cells from EPI-induced apoptosis, and reduced NFAT4 nuclear translocation and hypertrophy. This study suggests that EPI may affect SOCE in two phases: the initial enhancement phase and the following cell compensatory reduction phase. Administration of a SOCE blocker at the initial enhancement phase may protect cardiomyocytes from EPI-induced toxicity and hypertrophy.

## 1. Introduction

Anthracyclines listed in the 22nd (the latest) version of the World Health Organization (WHO) model list of essential medicines are among the most efficacious and widely used chemotherapy drugs since the late 1960s [1]. Epirubicin (EPI) belongs to the anthracycline family; it is often used together with new generation targeted drugs and play a major role in the modern era of cancer treatment. EPI kills cancer cells likely via multiple mechanisms, including DNA adduct formation, reactive oxygen species (ROS) production, and lipid peroxidation. While EPI makes a great contribution to the improvement of treatment outcomes, dose-limiting cardiotoxicity hinders its clinical application and often leads to requirements for regimen modification or even discontinuation [2]. The anthracycline-induced cardiotoxicity can be manifested either acutely during the treatment period or chronically, from several weeks to even years after treatment has stopped [3]. The associated cardiac dysfunction has a broad range of symptoms including cardiac hypertrophy, cardiomyopathy, and ultimately congestive heart failure [4].

Cardiac hypertrophy is the enlargement of the heart, which can be divided into two categories: physiological and pathological, both of which develop as an adaptive response to cardiac stress, but their underlying molecular mechanisms, cardiac phenotype and prognosis are distinctly different. For example, Ca^2+^ signaling-related genes are only changed in pathological hypertrophy but not in physiological hypertrophy [5]. Studies have revealed that intracellular Ca^2+^ regulates the calcineurin–NFAT signaling pathway and thus initiating hypertrophy-related gene transcription [6,7,8,9]. An increase in intracellular Ca^2+^ leads to the activation of the phosphatase activity of calcineurin, the dephosphorylation of NFAT family members, and their translocation to the nucleus to initiate gene transcription [6].

Store-operated Ca^2+^ entry (SOCE) is a ubiquitous Ca^2+^ entry pathway activated in response to the depletion of sarcoplasmic or endoplasmic reticulum (SR/ER) Ca^2+^ stores. Although SOCE has been well-studied in non-excitable cells and skeletal muscles, the understanding of its important role in cardiomyocytes is emerging [10,11]. SOCE machinery components, including stromal interaction molecule 1 (Stim1) as an ER Ca^2+^ sensor and Orais and transient receptor potential channels (TRPCs) as plasma membrane Ca^2+^ channels, have been shown to be essential for heart development and to regulate heart remodeling after stress [12]. Accumulating evidence also shows enhanced SOCE in cardiac hypertrophy and heart failure [7,8,9]. While dysregulated Ca^2+^ signaling has been reported to contribute to EPI-induced cardiotoxicity [13,14,15], whether SOCE plays a role in this process and in the consequent cardiac remodeling remains unknown. Thus, the objective of the present study is to determine the specific role of SOCE in EPI-induced cell apoptosis and hypertrophy in cardiomyocytes.

## 2. Materials and Methods

### 2.1. Chemicals and Reagents

Claycomb cell culture medium was purchased from Sigma-Aldrich. FBS (fetal bovine serum), PBS (phosphate-buffered saline), HBSS (Hanks’ balanced salt solution), and penicillin/streptomycin antibiotic were purchased from Invitrogen/Thermofisher Scientific Pittsburgh, PA, USA. Other reagents used include BTP2 and ML204 (Millipore Sigma, St. Louis, MO, USA), EPI (Alfa Aesar, Haverhill, MA, USA), thapsigargin (TG, Adipogen, San Diego, CA, USA), fura-2 AM (Biotium 50033, Fremont, CA, USA), DAPI (Invitrogen D357, Carlsbad, CA, USA), and phalloidin (Enzo BML-T111, New York, NY, USA).

### 2.2. Cell Culture

HL-1 cardiomyocytes were maintained in Claycomb medium supplemented with 10% FBS, 100 U/mL penicillin, 100 ug/mL streptomycin, 0.1 mM norepinephrine, and 2 mM L-glutamine [16,17]. HL-1 cells were cultured at 37 °C in a humidified 5% CO_2_ incubator.

### 2.3. Measurement of Intracellular Ca^2+^ Concentration

Intracellular Ca^2+^ concentrations in the HL-1 cell line was measured following previously published procedures [17]. In brief, the intracellular Ca^2+^ was measured using a fluorescence microscope with a SuperFluo 40× objective (N.A. 1.3) connected to a dual-wavelength spectrofluorometer (Horiba Photon Technology International, Piscataway, NJ, USA). The excitation wavelengths were set at 350 nm and 385 nm and the emission wavelength was set at 510 nm. Cells were loaded with 5 μM fura-2 acetoxymethyl ester (Biotium, Fremont, CA, USA) for 30 min at 37 °C in the dark. Cellular endoplasmic reticulum (ER) Ca^2+^ stores were depleted by 10 μM TG in 0.5 mM EGTA dissolved in balanced salt solution (140 mM NaCl, 2.8 mM KCl, 2 mM MgCl_2_, 10 mM HEPES, pH 7.2). SOCE was observed upon the rapid exchange of extracellular solution to bath saline containing 2 mM CaCl_2_ at indicated time. The intracellular Ca^2+^ elevation was presented as ΔF350 nm/F385 nm.

### 2.4. Cytotoxicity Assay

HL-1 cells were seeded at 1.5 × 10^5^ cells per well in a 29 mm glass-bottom dish. The cells were treated with vehicle, 20 µM BTP2, 1 µM EPI, or 20 µM BTP2 plus 1 µM EPI for 5 h. Then, the culture medium was removed, and cells were fixed with 4% paraformaldehyde for 10 min at room temperature. The paraformaldehyde was removed and then cells were immersed in 0.1% Triton X-100 in PBS for 10 min, washed with PBS twice, followed by incubation with PBS containing 6.6 µM phalloidin (Enzo, New York, NY, USA) for 15 min. The cells were washed with PBS three times, and then counter staining with PBS containing 1 µg/mL DAPI (1:500) for 5 min at room temperature in the dark. The cells were then washed with PBS twice and immersed in ProlongTM Gold antifade reagent (Life Technologies Corporate, Eugene, OR, USA). The fluorescence signals were observed using a DMi8 inverted microscope (Leica, Wetzlar, Germany) with a 40× objective (NA 1.3). The excitation/emission wavelengths set for DAPI and phalloidin were 405/430 nm and 547/572 nm, respectively. The imaging was performed at room temperature.

### 2.5. Western Blotting Analysis

HL-1 cardiomyocytes were lysed in modified RIPA buffer (150 mM NaCl, 50 mM Tris-Cl, 1 mM EGTA, 1% Triton X-100, 0.1% SDS, and 1% sodium deoxycholate, pH 8.0) containing protease inhibitors cocktail (Sigma-Aldrich, US) as previously described [18,19]. Protein concentration was quantified using a BCA kit (ThermoFisher, Pittsburgh, PA, USA). Equal amounts of proteins were loaded onto SDS polyacrylamide gels, and the separated proteins were transferred to PVDF membranes (Bio-Rad, Hercules, CA, USA). The blot was incubated with 5% non-fat dry milk blocking buffer (Bio-Rad, Hercules, CA, USA) for 1 h at room temperature and probed with specific primary antibodies in blocking buffer at 4 °C overnight. The primary antibodies used in this study included anti-caspase-3 (1:1000, catalog #9662, Cell Signaling Technology, Massachusetts, MA, USA) and anti-GAPDH (1:1000, GeneTex, Irvine, CA, USA). The next day, the blots were washed with PBST three times followed by incubation with secondary antibodies including the appropriate horse radish peroxidase (HRP)-conjugated goat anti-rabbit IgG (1:5000, Cell Signaling Technology, Massachusetts, USA) and anti-mouse IgG (1:5000, Cell Signaling Technology, USA). Signals were detected using the ECL detection method on a ChemiDoc instrument.

### 2.6. Cell Size Measurement

HL-1 cells seeded at 1 × 10^6^ cells per well in a 6-well plate were treated with vehicle or 20 µM BTP2, 1 µM EPI, or 20 µM BTP2 plus 1 µM EPI for 5 h followed by switching to normal culture media for 24 h. The cells were then observed and phase contrast imaging was conducted using a DMi8 inverted microscope (Leica, Wetzlar, Germany). The cell surface area was quantified using ImageJ and Graphpad 6 software.

### 2.7. Quantitative Reverse Transcription Polymerase Chain Reaction (qRT-PCR)

Total RNAs were extracted from HL-1 cells using Illustra RNAspin MiniRNA Isolation Kit and the quality and concentration of RNA were evaluated by photometrical measurement of 260/280 nm. The primers were obtained from Sigma Aldrich. Four hundred nanograms of total RNA was applied for reverse transcription using the qScript microRNA Synthesis Kit (QuantaBio, Beverly, MA, USA) following the manufacturer’s protocol. cDNA was diluted 1:5 in DNase-, RNase-, and protease-free water and 2 μL template was used for PCR. The primer pairs for BNP and GAPDH were used. The sequences for the BNP primers are forward (5′–3′) GCCAGTCTCCAGAGCAATTC and reverse (5′–3′) TCTTTTGTGAGGCCTTGGTC. The sequences for the GAPDH primers are forward (5′–3′) AGGTCGGTGTGAACGGATTTG and reverse (5′–3′) TGTAGACCATGTAGTTGAGGTCA. For qRT-PCR, QuantaBio PerfecTa SYBR Green FastMix ROX was used according to the manufacturer’s procedure. The signals generated by integration of SYBR Green into the amplified DNA were detected in a real-time machine (StepOne Plus Real-Time PCR System, ThermoFisher Scientific, USA). Data were expressed as 2-ΔΔCT relative to GAPDH gene expression.

### 2.8. Immunofluorescence Staining

Cells were seeded into 29 mm glass-bottom dishes. The cells were fixed with 4% paraformaldehyde for 10 min at room temperature. The paraformaldehyde was then removed and the cells were immersed in PBS containing 0.1% Triton X-100 for 10 min. After washing with PBS three times, the cells were blocked in PBS containing 0.1% Triton X-100 supplemented with 10% horse serum for 30 min at room temperature. Then, the cells were incubated with rabbit anti-NFAT4 primary antibody (1:100, ProteinTech, Rosemont, IL, USA) in blocking solution at 4 °C overnight. The next day, the cells were taken out and washed with PBS three times, then incubated with Alexa Fluor 488-labelled secondary antibody (1:500, Abcam, Cambridge, MA, USA) at room temperature in the dark for 1 h to visualize the expression and localization of NFAT4. The cells were counter-stained with PBS containing 1 μg/mL DAPI (1:500) for 5 min at room temperature in the dark and then immersed in ProlongTM Gold antifade reagent (Life Technologies Corporate, Eugene, OR, USA). Images were taken using a Nikon A1R HD25 LSM confocal microscope with a 40× oil immersion objective (NA 1.3) using GFP and DAPI filters (Ex: 488/405; Em: 509/430 nm).

### 2.9. RNA-Seq Data Analysis

The RNA-Seq dataset GSE217421 was used [20]. Different human induced pluripotent stem cell (iPSC)-derived cardiomyocyte cell lines were treated with 2 µM EPI or DMSP for 48 h, followed by bulk RNA-seq analysis. Differentially expressed gene were identified between drug- and control treated cell lines. A total 17 EPI samples and 56 control samples covering five different cell types were used with each cell type having a different number of replicates as shown in Appendix A. Appendix A shows all the control cell lines and replicate numbers. Two-way ANOVA of the effects of treatment (EPI vs. Con) and cell lines (five cell lines) was used for analysis.

### 2.10. Statistical Analysis

Data were analyzed using Graphpad Prism 6 software (Boston, MA, USA) unless indicated otherwise. The results were presented as mean ± standard deviation (SD) or as otherwise indicated. Comparisons between two groups were analyzed using a Student’s t-test. Comparisons among more than two groups were analyzed using one-way analysis of variance (ANOVA) followed by Bonferroni post hoc analysis. A *p* value of <0.05 was considered statistically significant in all experiments except the RNA-seq data analysis.

## 3. Results

### 3.1. SOCE Machinery Genes Were Downregulated by EPI Treatment in Human iPSC-Derived Cardiomyocytes

RNA-seq data analysis of human iPSC-derived cardiomyocytes showed that cells treated with 2 µM EPI for 48 h had significantly reduced expression of SOCE machinery genes, i.e., Orai1, Orai3, TRPC3, TRPC4, Stim1, and Stim2, and increased expression of TRPC2 (Figure 1A, Appendix A). The expression of Orai2, TRPC1, TRPC5, and TRPC6 were similar between the EPI and control groups.

To confirm the changes in SOCE in live cells, HL-1, a cardiomyocyte cell line derived from adult mouse atria was used for its easy culture and well-characterized cardiomyocyte properties. After being treated with 1 µM EPI or vehicle control (0.1% DMSO) for 6 h, HL-1 cells were loaded with 5 µM fluorescent Ca^2+^ indicator fura-2 AM at 37 °C in the dark for 30 min. The ER Ca^2+^ stores were depleted by 10 μM TG in BSS containing 0.5 mM EGTA. When re-introducing BSS containing 2 mM CaCl2, the intracellular Ca^2+^ level (presented as F350 nm/F385 nm) was monitored using live cell imaging and the SOCE was calculated as the difference (ΔF350/F385) between the peak and baseline before the addition of 2 mM Ca^2+^. Compared to vehicle control (black curve), SOCE was significantly reduced in the EPI-treated HL-1 cells (red curve) (Figure 1B,C).

### 3.2. Acute Treatment of EPI Increased SOCE in HL-1 Cardiomyocytes

In addition to transcriptional regulation, EPI can increase ROS production and lipid peroxidation. Oxidative stress has been shown to promote STIM1 oligomerization and alter channel activity [21]. We next examined whether acute treatment of EPI and its resulting oxidative stress can affect SOCE in HL-1 cells. Administration of BTP2, a SOCE inhibitor, significantly decreased ΔF350/F385 (0.142 ± 0.064) compared with that of the vehicle-treated control cells (0.188 ± 0.058, *n* = 35; ** *p* = 0.0051). This data confirmed the presence of BTP2-sensitive SOCE in HL-1 cardiomyocytes (Figure 2A,B,E). Contrary to prolonged treatment, acute treatment of EPI for only 30 min resulted in significantly enhanced SOCE in HL-1 cells (0.254 ± 0.069, *n* = 37) compared to those treated with the vehicle control (0.188 ± 0.058, *n* = 37; **** *p* < 0.0001). Addition of BTP2 could significantly decrease SOCE (0.045 ± 0.027, *n* = 36) in EPI-treated HL-1 cells compared to those treated with EPI alone (0.254 ± 0.069, *n* = 36; **** *p* < 0.0001), indicating that pharmacologically inhibiting SOCE with BTP2 can reduce the EPI-enhanced SOCE in HL-1 cells. Furthermore, ML204, a relative specific TRPC4 inhibitor could significantly reduce SOCE in HL-1 cells as well (Appendix A).

### 3.3. BTP2 Diminished EPI-Induced ROS Production in HL-1 Cardiomyocytes

The reciprocal regulation between mitochondria and intracellular Ca^2+^ suggests that SOCE may regulate mitochondrial ROS production. Thus, ROS were measured by using DHE dye in HL-1 cells treated with EPI (Figure 3). In the HL-1 cells treated with 1 μM EPI for 30 min, ROS levels were significantly increased compared to that in vehicle control cells. Interestingly, BTP2 was able to significantly inhibit ROS production in HL-1 cells even in the presence of EPI (Figure 3).

### 3.4. BTP2 Inhibited EPI-Induced Apoptosis in HL-1 Cardiomyocytes

Disruption of F-actin is a hallmark for apoptosis [22]. We next examined the expression of F-actin in HL-1 cells using phalloidin staining. Reduced expression of F-actin was observed in cells treated with 1 µM EPI for 5 h compared to that of vehicle-treated control cells (Figure 4A), suggesting that EPI induced apoptosis in HL-1 cardiomyocytes. When co-treated with 20 µM BTP2, the EPI-induced degradation of F-actin was partially rescued (Figure 4A,B), indicating that BTP2 inhibited EPI-induced F-actin disruption.

Anthracyclines have been shown to induce apoptosis in HL-1 cardiomyocytes through caspase-3 [23]. We then examined the levels of cleaved caspase-3 in HL-1 cells. EPI induced abundant amounts of cleaved caspase-3, evidenced by the Western blot analysis (Figure 4C,D). The EPI-increased level of cleaved caspase-3 was significantly diminished by co-treatment with 20 µM BTP2. Consistent with the data from F-actin degradation, the cleaved caspase-3 analysis again indicated that EPI induced apoptosis in HL-1 cardiomyocytes, which could be alleviated by BTP2.

### 3.5. BTP2 Inhibited EPI-Induced Hypertrophy in HL-1 Cardiomyocytes

EPI-induced cardiac remodeling includes hypertrophy. SOCE plays a major role in pathophysiological hypertrophy. We thus examined whether BTP2 can inhibit EPI-induced hypertrophy in HL-1 cardiomyocytes. HL-1 cells were treated with vehicle control, 1 µM EPI, or co-treated with 1 µM EPI and 20 µM BTP2 for 5 h, followed by drug withdrawal and then growth in normal culture medium for 24 h. Phase contrast images were then taken of these cells and the surface area of the HL-1 cardiomyocytes was measured and quantified. EPI treatment increased the size of cardiomyocytes to almost twice that of vehicle-treated control cardiomyocytes (Figure 5A,B). In the BTP2 and EPI co-treatment group, the size of HL-1 cells was significantly reduced compared to that in the EPI group.

The expression of brain natriuretic peptide (BNP), a specific marker of cardiac hypertrophy [22], was also examined. As shown in Figure 5C, the mRNA level of BNP was significantly increased upon the treatment with 1 µM EPI (4.861 ± 0.697, *n* = 9) compared to that of vehicle-treated cells (control, 1.010 ± 0.155, *n* = 9). Consistent with the cell size analysis, BTP2 could significantly alleviate EPI-induced BNP expression (3.054 ± 0.260) in HL-1 cells.

These data indicate that blocking SOCE by BTP2 can reduce EPI-induced hypertrophy in HL-1 cardiomyocytes.

### 3.6. BTP2 Inhibited EPI-Induced NFAT4 Nuclear Translocation in HL-1 Cardiomyocytes

Nuclear factor of activated T cells (NFAT) was reported to be a critical nuclear transcriptional factor regulating cardiac hypertrophy [24]. We lastly examined whether SOCE contributes to EPI-induced hypertrophy through the NFAT pathway in HL-1 cells. Since NFAT4 is the most abundant one out of the five subtypes of NFAT expressed in cardiomyocytes [25], we focused on NFAT4 in this study. After being treated with vehicle, 1 µM EPI, 20 µM BTP2, or 1 µM EPI combined with 20 µM BTP2 for 5 h, HL-1 cells were cultured in growth media for another 24 h until fixation and immunostaining with anti-NFAT4 antibody. HL-1 cells treated with 10 µM ionomycin for 15 min were used as a positive control for NFAT4 immunostaining since ionomycin is a strong activator for NFAT signaling [26]. The nuclear translocation of NFAT4 was examined by confocal microscopy imaging. As shown in Figure 6, EPI treatment induced NFAT4 nuclear translocation (indicated by the white arrows), whereas co-treatment with BTP2 showed minimal NFAT4 nuclear translocation. This data suggested that the EPI-induced nuclear translocation of NFAT4 was inhibited by BTP2 in HL-1 cells.

## 4. Discussion

EPI is a widely used anthracycline chemotherapy drug, but it also causes cardiotoxicity and results in heart remodeling and even failure. This study confirmed that EPI can induce ROS production, cell apoptosis, and hypertrophy in cardiomyocytes. Furthermore, this study showed that acute treatment of EPI can increase SOCE in HL-1 cells and blocking SOCE by BTP-2 not only reduced EPI-enhanced SOCE (Figure 2), but also alleviated EPI-induced apoptosis (Figure 4) and hypertrophy (Figure 5). Although SOCE has been associated with hypertrophy in cardiomyocytes and heart failure, this study provides the first evidence, to our knowledge, that SOCE plays a key role in EPI-induced cardiotoxicity and hypertrophy. More importantly, this study may shed light on developing an approach to alleviate EPI-induced cardiotoxicity by targeting SOCE in the initial phase of EPI treatment (working model is shown in Figure 7).

It is well-known that SOCE has a complex nature and co-exists with other Ca^2+^ influx mechanisms, such as receptor-operated Ca^2+^ entry (ROCE). SOCE machinery may contain several molecules as channel complexes at the plasma membrane interacting with STIMs at the SR/ER. Previous reports showed that Orai1 is expressed in HL-1 cells and knockdown of Orai1 could abolish SOCE in HL-1 cells [27]. In addition, TRPC1, 3/6, and 4 may also form SOCE channel complexes in hypertrophic cardiomyocytes [24] and STIM1 can bind and regulate TRPC1, TRPC4, and TRPC5 [28]. The current study showed evidence for a bona fide, BTP2-sensitive SOCE in HL-1 cells. Since BTP2 can block both Orai [29] and TRPC channels [30], our current data cannot exactly pinpoint whether Orais or TRPCs mediate SOCE in hypertrophic cardiomyocytes. RNA-seq analysis showed that treatment of EPI significantly reduced Orai1, Orai3, TRPC3, and TRPC4 expression in human iPSC-derived cardiomyocytes, which is consistent with reduced SOCE (Figure 1A, Appendix A). Interestingly, ML204, a relatively selective blocker of TRPC4 could significantly reduce SOCE in HL-1 cells (Appendix A). These data suggest that these Orais and TRPCs may contribute to SOCE in cardiomyocytes. Future investigation is required to dissect the exact components in the SOCE channel complex in cardiomyocytes, which contribute to EPI-induced cardiotoxicity.

After cardiomyocytes survived the cardiotoxicity after EPI treatment, they may undergo cell remodeling which leads to hypertrophy, cardiac remodeling, and eventual heart failure. SOCE plays a major role in the pathogenesis of heart hypertrophy. Numerous studies suggest that pathological stimuli activate SOCE and further trigger the NFAT signaling cascade, which is critical for the regulation of growth gene expression and promotion of cardiomyocyte hypertrophy. Suppression of SOCE machinery genes, such as STIM1 and Orai1, attenuates the hypertrophic responses to pressure overload or agonists [31,32]. Our current findings are in line with these previous reports, indicating that EPI-induced cardiomyocyte hypertrophy could also be inhibited by SOCE blocker.

Interestingly, RNA-seq data analysis of human iPSC-derived cardiomyocytes showed that cells treated with 2 µM EPI for 48 h had significantly reduced expression of SOCE machinery genes, e.g., Orai1, Orai3, TRPC3, TRPC4, Stim1, and Stim2 (Figure 1). Intracellular Ca^2+^ measurement in live HL-1 cells confirmed that SOCE was indeed reduced in cardiomyocytes treated with EPI for 6 h (Figure 1B,C) or longer. The apparent discrepancy suggests that EPI may affect SOCE in two phases: the initial enhancement phase followed by cell compensatory reduction phase. The initial enhancement phase is likely due to immediately increased ROS production and lipid peroxidation right after administration of EPI. The rapid generation of ROS has been best studied in myocardial ischemia–reperfusion models [33]. In addition to the regulatory roles of ROS in many cellular events [34], oxidative stress is also able to promote STIM1 oligomerization, deplete ER Ca^2+^, and active SOCE [21]. Since there is a reciprocal regulation between mitochondria and SOCE, EPI-triggered initial mitochondrial ROS production could be further amplified by enhanced SOCE, which is supported by the evidence that blocking SOCE by BTP2 attenuated EPI-triggered ROS production (Figure 3). Additionally, ROS is also known to directly activate TRPCs channels [35,36]. During the initial enhancement phase, EPI triggers the apoptotic pathway in cardiomyocytes. The surviving cardiomyocytes from the initial phase may develop compensatory mechanisms at the transcription level. This may explain why prolonged treatment of EPI (at 48 h) resulted in a reduction in the expression of SOCE machinery genes.

Chemotherapeutic agents (anthracycline therapy in particular) have been reported to damage the F-actin of cells. In cardiac H9c2 cells, doxorubicin reduces number of F-actin filaments, especially at higher concentrations [37]. The reorganization of F-actin filaments and characteristic features of apoptosis have also been reported in Chinese hamster ovary cells, pancreatic β cells, breast cancer cells, and other cells upon doxorubicin treatment [38,39,40].

Others and our previous studies suggest that SOCE is an effective chemotherapy drug target [18,19,41,42,43]. The findings of the present study have shown that SOCE contributes to EPI-induced cardiotoxicity, indicating that SOCE blockers may be able to protect cardiomyocytes from the side effects of anthracycline chemotherapy drugs. Together, the results suggest that SOCE blockers may be dual-function drugs for both chemotherapy and cardio-protection.

## Figures and Tables

**Figure 1 cells-12-00723-f001:**
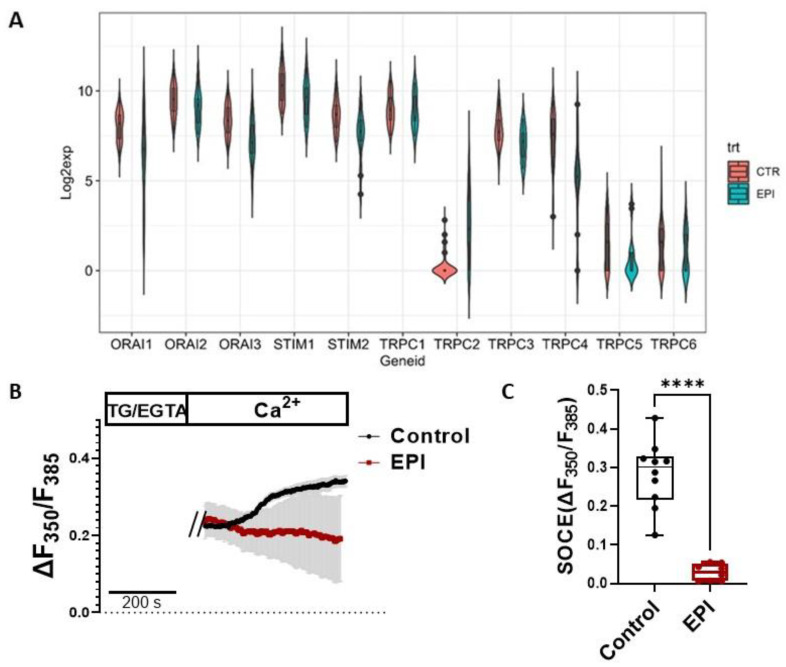
SOCE machinery genes were downregulated by EPI in cardiomyocytes. (**A**) RNA-seq data analysis of human iPSC-derived cardiomyocytes treated with 2 µM EPI for 48 h. SOCE machinery genes include Orais, TRPCs, and Stims. (**B**) Representative traces of intracellular Ca^2+^ in HL-1 cells treated with vehicle control or 1 µM EPI for 6 h. (**C**) Statistics of changes in intracellular Ca^2+^ in HL-1 cells without (0.28 ± 0.085) or with the treatment of EPI (0.02897 ± 0.021). Mean ± SD, *n* = 10, ****: *p* < 0.0001 (based on *t*-test).

**Figure 2 cells-12-00723-f002:**
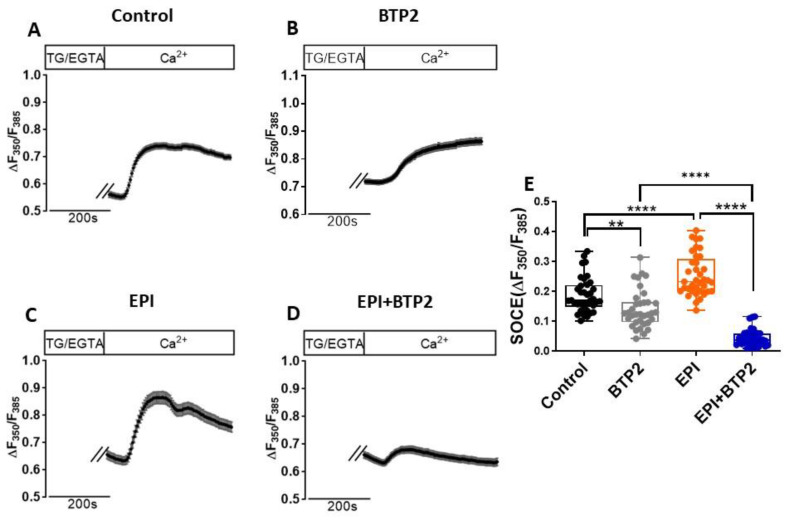
Acute treatment of EPI increased SOCE in HL-1 cardiomyocytes. Ratio of fluorescence of fura-2 AM at two excitation wavelengths (F350 nm/F385 nm) was used to monitor changes in intracellular Ca^2+^ concentration. (**A**–**D**) Representative traces of intracellular Ca^2+^ in HL-1 cells treated with vehicle control, 20 µM BTP2, 1 µM EPI, or 20 µM BTP2 plus 1 µM EPI for 30 min. (**E**) Statistical analysis of SOCE in HL-1 cells. Mean ± SD, *n* = 35. **: *p* = 0.0056, ****: *p* < 0.0001 (based on one-way ANOVA and Bonferroni post hoc analysis).

**Figure 3 cells-12-00723-f003:**
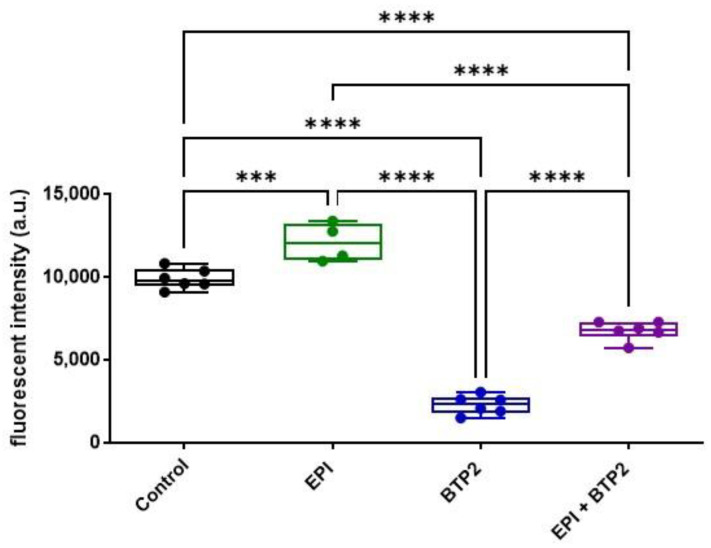
EPI induced ROS production in HL-1 cardiomyocytes. ROS was measured in HL-1 cells by using DHE dye in control group with only resting cells; 1 µM EPI treatment for 30 min (EPI); 20 µM BTP2 (BTP2) treatment; 1 μM EPI together with 20 µM BTP2 treatment for 30 min (EPI + BTP-2). Quantitative fluorescent intensity (a.u.) from each independent well was showed. *n* = 6, Mean ± SD. ***: *p* < 0.001; ****: *p* < 0.0001. (Based on One Way ANOVA and Bonferroni post hoc analysis).

**Figure 4 cells-12-00723-f004:**
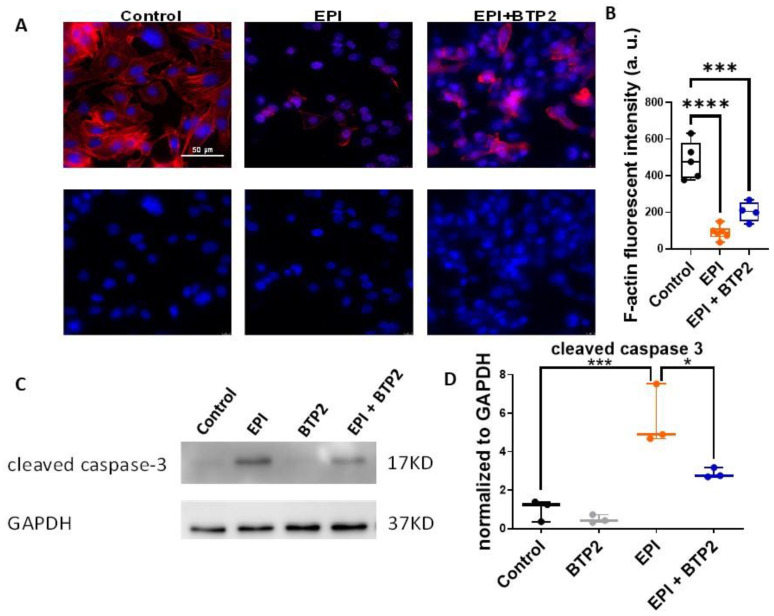
BTP2 inhibited EPI-induced apoptosis in HL-1 cardiomyocytes. (**A**) Cell nuclei were stained with DAPI (blue) and F-actin was stained with phalloidin (red). F-actin degradation was observed with 1 µM EPI treatment for 5 h. With 20 µM BTP2 co-treatment, the EPI-induced F-actin disruption was partially rescued. (**B**) The F-actin fluorescence intensity was quantified. ***: *p* < 0.001; ****: *p* < 0.0001 (**C**) Western blotting analysis of cleaved caspase 3 expression in HL-1 cardiomyocytes. Cells were treated by vehicle (control), 1 µM EPI, 20 µM BTP2, or 1 µM EPI plus 20 µM BTP2 for 5 h followed by normal culture conditions for 24 h. GAPDH was used as loading control. (**D**) The quantification of cleaved caspase 3 normalized to the expression of GAPDH. Three independent biological replicates were carried out and used for the quantification. ***: EPI vs. Control, *p* = 0.0009; *: EPI + BTP2 vs. EPI, *p* = 0.0218 (based on one-way ANOVA and Bonferroni post hoc analysis).

**Figure 5 cells-12-00723-f005:**
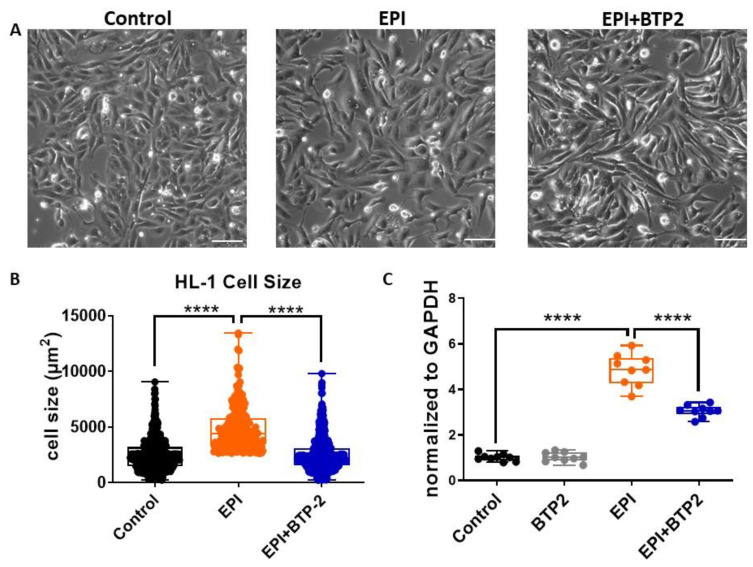
BTP2 inhibited EPI-induced hypertrophy in HL-1 cardiomyocytes. HL-1 cells treated with 1 µM EPI for 5 h followed by fresh media culture for 24 h. (**A**) Phase contrast images of HL-1 cells treated with 1 µM EPI (EPI), 1 μM EPI together with 20 µM BTP2 (EPI + BTP-2), or vehicle-treated control (control). Scale bar, 50 µm. (**B**) Quantification of the cell surface area of HL-1 cardiomyocytes. Mean ± SEM, n ≥ 581 per group. ****: *p* < 0.0001 (based on one-way ANOVA and Bonferroni post hoc analysis). (**C**) BTP2 reduced hypertrophy marker BNP transcript in HL-1 cardiomyocytes. Quantitative reverse transcription PCR expression levels of BNP were normalized to GAPDH and plotted relative to the level in the vehicle-treated control cells. *n* = 9, triplicates from three independent experiment. Mean ± SD. ****: *p* < 0.0001 (based on one-way ANOVA and Bonferroni post hoc analysis).

**Figure 6 cells-12-00723-f006:**
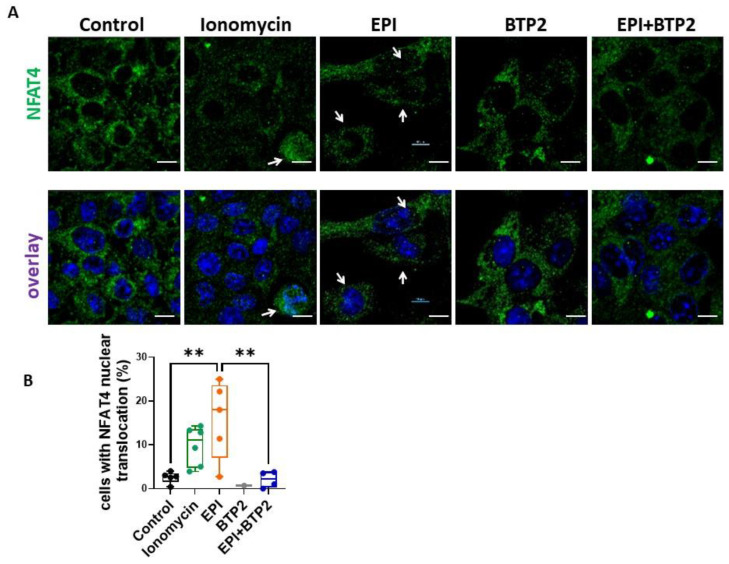
BTP2 inhibited the EPI-induced nuclear translocation of NFAT4 in HL-1 cardiomyocytes. HL-1 cells were treated with vehicle, 1 µM EPI, 20 µM BTP2, or 1 µM EPI combined with 20 µM BTP2 for 5 h, followed by drug withdrawal and culture for 24 h and stained for NFAT4. Cells treated with 10 µM ionomycin for 15 min were used as positive control. (**A**) Representative confocal images of NFAT. Scale bar, 10 µm. white arrows indicate the co-localization of NFAT4 and Nuclei (**B**) Statistical analysis of the percentage of cells with NFAT nuclear translocation. Each dot is an averaged datapoint from a single confocal image. Each image represents one biological replicate. Mean ± SD. **: *p* < 0.01 (based on one-way ANOVA and Bonferroni post hoc analysis).

**Figure 7 cells-12-00723-f007:**
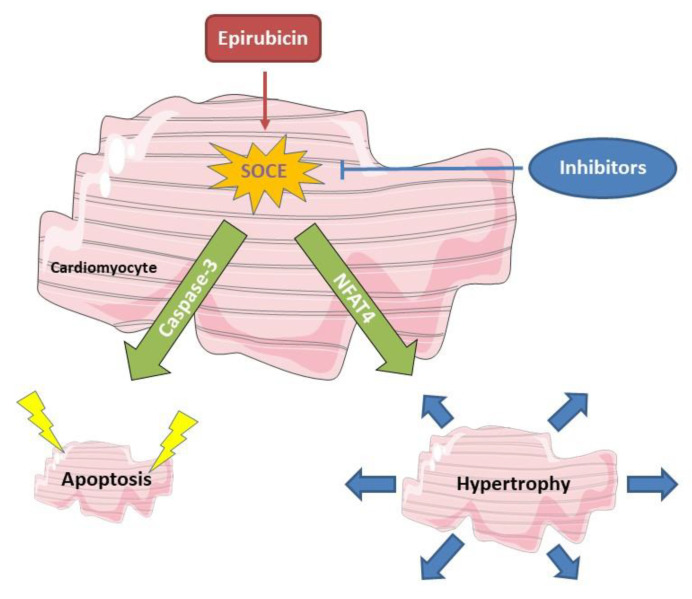
Working model. Acute treatment of EPI enhanced SOCE, triggering NFAT4-mediated hypertrophy and caspase3-mediated apoptosis in cardiomyocytes. Both the EPI-induced apoptosis and hypertrophy can be inhibited by blocking SOCE.

## Data Availability

Publicly available datasets were analyzed in this study. This data can be found here: GSE217421 (https://www.ncbi.nlm.nih.gov/geo/query/acc.cgi?acc=GSE217421, accessed on 30 November 2022.)

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
