# Peer review of "Blocking Store-Operated Ca2+ Entry to Protect HL-1 Cardiomyocytes from Epirubicin-Induced Cardiotoxicity"

_cells, 2023, doi:10.3390/cells12050723_

Round 1

Reviewer 1 Report

Anthracyclines have long been a staple of chemotherapy and are currently used extensively in the fight against cancer. Their cardiotoxicity, however, continues to be a serious barrier to their practical implementation.

The authors give evidence for a function for store-operated calcium entry in cardiotoxicity caused by epirubicin in the current study that has been submitted to Cells.

Although the topic is interesting to journal readers, there are a number of significant issues that need to be resolved.

Major issues:

-        A major drawback of the study is the use of a non-specific ORAI blocker. BTP2 has long been shown as both a CRAC (ORAIs) and a ROCE (TRPC3/6) blocker (PMID: 25128576; PMID: 20448219; PMID: 20410214; PMID: 15647288) at even much lower concentrations. The use of more specific ORAI1 blockers (Pyr6, syntax-66, GSK7975A, or JPIII) is recommended at least for the Ca2+/ROS imaging experiments. Additionally, the authors must rule out a potential function for TRPC3/6 channels in the Ca2+-mediated epirubicin-induced cardiotoxicity as these channels have been linked to doxorubicin-induced cardiotoxicity (PMID: 28495616; PMID: 31241172; PMID: 35096991).

Testing ORAI1 knockdown (siRNA) in the Ca2+/ROS experiments is also crucial to further prove its role in epirubicin-induced cardiotoxicity.

-        Mechanistic understanding of how epirubicin activates ORAI1 is lacking in the study. The authors stated that ROS could be the triggering factor, with a positive feedback loop existing between SOCE and ROS. This remains speculative unless experimentally proven; live-cell Ca2+ imaging should be performed with ROS scavengers to check whether SOCE inhibition would result or not. Furthermore, endoplasmic reticulum stores have to be assessed for Ca2+ depletion which might explain SOCE activation. Since the authors used thapsigargin in their protocol, quantifications of the Ca2+ peaks resulting from TG should be presented to assess the internal stores. Hence, the omitted Ca2+ curve parts in figures 1 and 2 should be shown.

-        What is the rationale for using several EPI treatment protocols, 10 μM EPI for 5 hrs for the cytotoxicity tests (line 103) (erroneously stated as 1 μM EPI in the results section, line 242); 1 μM EPI for 2 hrs for cell size measurement (erroneously stated as 1 μM EPI for 4 hrs in the results section, line 269); 2 μM EPI for 48 hours for RNA-seq; 1 μM EPI for 24 hours for the caspase western blot; 1 μM EPI for 6 hours or 30 minutes for Ca2+ imaging? A schematic representation of the different protocols and rationales would be of great value for clarification. Nevertheless, these different regimens weaken the drawn conclusions related to SOCE timeline activation by EPI.

-        Please be precise about what were the control (vehicle) experiments. Since DMSO is used to dissolve BTP2, was it used as a negative control?

-        Please provide higher magnification photos for the NFAT4 immunofluorescence, so the nuclear localization of the protein would be better shown.

-        In statistical analysis, were all data following normal distribution? If not, Kruskal-Wallis ANOVA followed by Dunn’s test should be used in skewed data. Besides, Tukey’s test was stated in the methods section, whereas Bonferroni’s was stated in figure legends.

-        Why is the decrease in Ca2+ response after BTP2 treatment more important in cells treated with EPI in comparison to control cells (Figure 2E)? Shouldn’t both levels be comparable (grey and blue bars)?

-        Does blocking ORAI1 with BTP2 relate to the decrease in its expression after 48 hours? To strengthen the drawn conclusion about the two SOCE phases, ORAIs expression (Figure 1A) should be studied under BTP2 treatment too.

Minor issues:

-        Why was glucose omitted from the basal saline solution used in the imaging experiments? Won’t this stress the cells further?

-        Lines 72-73: The references 13-15 do not relate to Ca2+-mediated epirubicin-induced cardiotoxicity, but rather to doxorubicin.

-        Lines 340-341: these Ca2+ imaging results were not reported in any of the figures.

-        Lines 22-26: Please correct the English transition since it appears as if the RNA-seq was performed in a previous study and not the current one.

English typos:

-        Line 31: “treatment by BTP2”

-        Line 190: “After being treated with 1 μM EPI”

-        Line 193: “When re-introducing BSS”

-        Line 231: “BTP2 was able significantly to inhibit ROS”

-        Line 276: “a specific marker of cardiac hypertrophy”

-        Line 298: “we focused on NFAT4 in our study”; “After being treated with vehicle”

-        Line 331: “activate SOCE”

Reviewer 2 Report

Comments to the authors:

Epirubicin (EPI) is an anthracycline type chemotherapeutic drug having cardiotoxicity as doxorubicin has, and Fura-2 (a fluorescent dye) has a significant protection in EPI treated cardiomyocytes by the reduction of SOCE machinery gene expression.

The main aim of the study was to investigate the role of intracellular Ca2+ (SOCE) and ROS in EPI-induced cardiotoxicity in HL-1 mouse cell line. It was concluded that SOCE blocker (BTP2) may protect cardiomyocytes from EPI-induced toxicity.

The manuscript is well designed and written, however, this reviewer did not correct the typos throughout the manuscript.

Introduction: 41-42. line; there is a latest edition of the World Health Organization (WHO) model list of essential medicines; 22nd edition (2021).

  • 2.1. Chemicals and reagents: 79th line; "The abbreviations (FBS, PBS, HBSS) however commonly used, should be explained in the text. i.e. FBS (fetal bovine serum), PBS (phosphate buffered saline), HBSS (Hanks' balanced salt solution)."
  • 2.10. Statistical analysis: 178th line; "Comparisons among more than two groups..." should be used for better readability instead "Comparisons among >two groups..."
  • 3.1. SOCE machinery genes were downregulated by EPI treatment in human iPSC-derived cardiomyocytes: 187th line: "... and TRPCs remained..." should be written as "... TRPC1 and TRPC6 remained..." for easier understanding.
  • Minor criticism: "It is not necessary to use multiple asterisk (*) to show the different significance levels between the values, one asterisk is more than enough in each figure."

  • The following figures should be larger in the revised version:
    • Figure 1. B; C
    • Figure 2. E
    • Figure 4. B; C; D
    • Figure 5. B; C
    • Figure 6. B

Additional comments:

Although it is indicated by the policy of the Journal that latest (about 5 years back) publications should be cited in a submitted manuscript, however, this reviewer believes that some classic papers (see below) could be also acknowledged in the revision of this (Liu X et al.) manuscript.

Fura-2 was extensively studied in calcium signaling several decades ago, therefore, some of the following classic papers may be acknowledged and cited in the revised version:

-      Science. 1987 Jan 16;235(4786):325-8. doi: 10.1126/science.3798114

-      Am J Physiol. 1989 Jun;256(6 Pt 1):C1120-30. doi: 10.1152/ajpcell.1989.256.6.C1120

-      Eur J Pharmacol. 1994 Dec 12;271(1):1-8. doi: 10.1016/0014-2999(94)90257-7

-      Am J Physiol. 1996 Jul;271(1 Pt 1):C391-7. doi: 10.1152/ajpcell.1996.271.1.C391

-      Am J Physiol. 1996 Jul;271(1 Pt 1):C391-7. doi: 10.1152/ajpcell.1996.271.1.C391

Additionally, other classic papers, ROS (reactive oxygen species) was also DIRECTLY detected in stress-induced myocardial ischemia/reperfusion-induced injury, which substantially modify Ca2+ handling alone or together with the formation ROS. Thus, the classic papers below should be also acknowledged in the revised version of the manuscript (Liu X et al.).

-      Proc Natl Acad Sci U S A. 1987 Mar;84(5):1404-7. doi: 10.1073/pnas.84.5.1404;

-      Circ Res,. 1987 Nov;61(5):757-60.  doi: 10.1161/01.res.61.5.757

-      Am Heart J. 1990 Oct;120(4):819-30. doi: 10.1016/0002-8703(90)90197-6

-      J Magn Reson. 2021 Aug;329:107024. doi: 10.1016/j.jmr.2021.107024. Epub 2021 Jun 9

Finally:

It has been recently emphasized, in several publications, that an increase in intracellular calcium content and generation of ROS leads (as stressors) to various cellular signaling, modifying the functions of gen, ion channels, producing apoptotic, necrotic and autophagic cell deaths. Thus, some recent papers (see below) may be also acknowledged in the Introduction or the Discussion in the revised version (Liu X et al.).

-      X Z Han 1S GaoY N ChengY Z SunW LiuL L TangD M Ren, Biosci Trends, 2012 Feb;6(1):19-25. doi: 10.5582/bst.2012.v6.1.19

-      Haines DD, Juhasz B, Tosaki A. J Cell Mol Med. 2013 Aug;17(8):936-57. doi: 10.1111/jcmm.12074. Epub 2013 Jun 22

-      Senoner T, Dichtl W. Nutrients. 2019 Sep 4;11(9):2090. doi: 10.3390/nu11092090

-      Chu Y, Li L, Liu Y, Wu Y, Bai H, Liu J, Yuan X, Zhang Z. Pharmazie. 2020 Jul 1;75(7):335-338. doi: 10.1691/ph.2020.0427

-      Kihara M, Kaiya H, Hirai Y, Katayama H, Terao A, Nishikawa M.Peptides. 2021 Mar;137:170471. doi: 10.1016/j.peptides.2020.170471. Epub 2020 Dec 16

-      Zhang M, Qi J, He Q, Ma D, Li J, Chu X, Zuo S, Zhang Y, Li L, Chu L.Phytother Res. 2022 Sep;36(9):3619-3631. doi: 10.1002/ptr.7528. Epub 2022 Jun 23

-      Syahputra RA, Harahap U, Dalimunthe A, Nasution MP, Satria D.Molecules. 2022 Feb 15;27(4):1320. doi: 10.3390/molecules27041320

Round 2

Reviewer 1 Report

Dear Editors,

The authors have adequately addressed all the concerns. I have no further comments.

Minor issue: Unless I have missed it, but where did the authors use the GSK-7975A mentioned in line 84? No results seem to be related to this compound.

Kind Regards

Author Response

We thank this reviewer for your speedy review. We are pleased to know that we have addressed your previous concerns adequately.

Regarding the comment on “English language and style are fine/minor spell check required”, we have our manuscript checked by a native English-speaking colleague, Dr. Frank Yi.

Regarding the minor issue on “where did the authors use the GSK-7975A mentioned in line 84”, we are very sorry about the error. We did not have that result. The word GSK-7975A is removed from Line 84. All the changes can be tracked in Word File.
